# Transsynaptic modulation of presynaptic short-term plasticity in hippocampal mossy fiber synapses

David Vandael[1,2,3✉], Yuji Okamoto [1,3] & Peter Jonas [1✉]

The hippocampal mossy fiber synapse is a key synapse of the trisynaptic circuit. Post-tetanic potentiation (PTP) is the most powerful form of plasticity at this synaptic connection. It is widely believed that mossy fiber PTP is an entirely presynaptic phenomenon, implying that PTP induction is input-specific, and requires neither activity of multiple inputs nor stimulation of postsynaptic neurons. To directly test cooperativity and associativity, we made paired recordings between single mossy fiber terminals and postsynaptic CA3 pyramidal neurons in rat brain slices. By stimulating non-overlapping mossy fiber inputs converging onto single CA3 neurons, we confirm that PTP is input-specific and non-cooperative. Unexpectedly, mossy fiber PTP exhibits anti-associative induction properties. EPSCs show only minimal PTP after combined pre- and postsynaptic high-frequency stimulation with intact postsynaptic $Ca^{2+}$ signaling, but marked PTP in the absence of postsynaptic spiking and after suppression of postsynaptic $Ca^{2+}$ signaling (10 mM EGTA). PTP is largely recovered by inhibitors of voltage-gated R- and L-type $Ca^{2+}$ channels, group II mGluRs, and vacuolar-type $H^+$-ATPase, suggesting the involvement of retrograde vesicular glutamate signaling. Transsynaptic regulation of PTP extends the repertoire of synaptic computations, implementing a brake on mossy fiber detonation and a "smart teacher" function of hippocampal mossy fiber synapses.

[1] Cellular Neuroscience, IST Austria (Institute of Science and Technology Austria), Klosterneuburg, Austria. [2] Present address: Netherlands Institute for Neuroscience, Royal Netherlands Academy for Arts and Sciences (KNAW), Amsterdam, The Netherlands. [3] These authors contributed equally: David Vandael, Yuji Okamoto. ✉email: david.vandael@ist.ac.at; peter.jonas@ist.ac.at

The hippocampal mossy fiber synapse is a key synapse of the trisynaptic circuit of the hippocampus[1–5]. Post-tetanic potentiation (PTP) is the most powerful form of plasticity at this synaptic connection[3,6–9]. We previously found that PTP depends on vesicle pool refilling during its induction phase, leading to an increase in the size of the readily releasable pool (RRP)[10]. Moreover, extra vesicles can be stored for extended periods of time that match in vivo recorded inter-event intervals (IEIs) of active dentate granule cells[10]. Thus, PTP might be an important mechanism in the formation of short-term memory engrams. However, to define the computational power of this storage mechanism, the precise induction rules need to be determined.

It is widely believed that both induction and expression of mossy fiber PTP are entirely presynaptic[4,5]. This would imply that induction is synapse-specific, and requires neither cooperative activation of multiple inputs nor associative stimulation of postsynaptic neurons[11]. However, direct experimental mapping of the precise induction rules of mossy fiber PTP has been difficult, because single inputs cannot be reliably stimulated. To measure PTP induction rules at hippocampal mossy fiber synapses, we performed paired recordings between hippocampal mossy fiber terminals and postsynaptic CA3 pyramidal neurons. We found that mossy fiber PTP showed synapse specificity and lack of cooperativity, as predicted[4,5]. However, PTP not only lacked associativity[12–15], but rather showed anti-associative induction properties. Postsynaptically shaped induction rules may help to prevent excessive detonation, increasing the computational power of mossy fiber synapses in the hippocampal network. A preliminary account of the work has been published in preprint form Vandael et al.[16].

## Results

**Specificity and non-cooperativity of mossy fiber PTP.** Synapse specificity, cooperativity, and associativity are hallmark properties of induction of synaptic plasticity at glutamatergic synapses. However, rigorous testing of these properties is often difficult, requiring defined stimulation of individual synaptic inputs. To test specificity and cooperativity of PTP at the hippocampal mossy fiber synapse, we combined tight-seal cell-attached stimulation of single mossy fiber terminals[9,10], with interleaved extracellular stimulation of the mossy fiber tract (Fig. 1a–c) in rat brain slices. To measure excitatory postsynaptic currents (EPSCs) in CA3 pyramidal neurons, postsynaptic neurons were held under voltage-clamp conditions at $-70$ or $-80$ mV throughout the entire recording, initially using a K-gluconate internal solution with 10 mM of the $Ca^{2+}$ chelator ethylene glycol-bis(2-aminoethylether)-N,N,N′,N′-tetraacetic acid (EGTA; Fig. 1c, d).

To test cooperativity, we compared the magnitude of PTP between unitary EPSCs evoked by single-terminal stimulation and compound EPSCs evoked by axon tract stimulation. High-frequency stimulation with 100 stimuli at 100 Hz ($HFS_{100}$) applied to a single mossy fiber terminal induced robust PTP of mossy fiber terminal-evoked unitary EPSCs ($673.6 \pm 285.7\%$; $65.9 \pm 23.9$ pA in control conditions versus $455.7 \pm 153.8$ pA 20–60 s after $HFS_{100}$; 9 pairs; $P = 0.0179$; Fig. 1d–f). Similarly, $HFS_{100}$ applied to the mossy fiber tract induced reliable PTP of tract stimulation-evoked compound EPSCs ($411.6 \pm 113.6\%$; $249.8 \pm 92.3$ pA in control conditions versus $628.4 \pm 146.3$ pA 20–60 s after $HFS_{100}$; $P = 0.007$; Fig. 1d–f). EPSCs evoked by axon tract stimulation under control conditions had, on average, a 2.6-fold larger amplitude than EPSCs evoked by presynaptic terminal stimulation, verifying the stimulation of multiple synaptic inputs.

To test synapse specificity, we examined the effects of $HFS_{100}$ applied at the mossy fiber tract on EPSCs evoked by single-terminal stimulation. Independence of the two inputs was verified by continuous monitoring of action currents at the presynaptic terminal (Fig. 1c; 3 out of 12 recordings had to be excluded based on this criterion). $HFS_{100}$ applied to the mossy fiber tract resulted in an only minimal change of the amplitude of presynaptic terminal-evoked EPSCs ($108.8 \pm 19.6\%$; $118.7 \pm 36.6$ pA in control and $142.2 \pm 71.6$ pA 20–60 s after $HFS_{100}$; 9 pairs; $P = 0.779$; Fig. 1d–f). Similarly, $HFS_{100}$ applied to the presynaptic terminal left the amplitude of the mossy fiber tract-evoked EPSCs unchanged ($164.6 \pm 36.6\%$; $172.5 \pm 58.7$ pA in control and $265.2 \pm 105.8$ pA 20–60 s after $HFS_{100}$; 9 pairs; $P = 0.678$; Fig. 1d–f). Taken together, these results demonstrate that PTP is a synapse-specific phenomenon.

**Anti-associativity of mossy fiber PTP.** We next asked whether PTP induction was associative or not[12–14] (Fig. 2). To test the effects of postsynaptic spiking, presynaptic mossy fiber terminals were stimulated in the tight-seal cell-attached configuration, while the postsynaptic CA3 pyramidal neuron was held in the whole-cell current-clamp configuration during induction (Fig. 2). Since a rise in postsynaptic $Ca^{2+}$ might be required, we used a postsynaptic internal solution containing 0.1 mM EGTA in this series of experiments. Surprisingly, PTP evoked by presynaptic $HFS_{100}$ paired with postsynaptic $HFS_{100}$ in current-clamp mode, was absent ($129.7 \pm 17.9\%$; $566.5 \pm 112.1$ pA in control conditions versus $666.0 \pm 131.2$ pA after $HFS_{100}$; 13 pairs; $P = 0.22$; Fig. 2a–c). These results not only indicate that postsynaptic action potentials (APs) are not required for PTP induction, but further suggest that postsynaptic activity suppresses PTP induction, implying an anti-associative induction mechanism (Fig. 2a–c).

To test the possible role of a rise in postsynaptic $Ca^{2+}$, we repeated the associativity experiment (presynaptic $HFS_{100}$ paired with postsynaptic $HFS_{100}$) with a postsynaptic intracellular solution containing 10 mM of the $Ca^{2+}$ chelator EGTA. In the presence of 10 mM postsynaptic EGTA, pre-postsynaptic $HFS_{100}$ led to a marked increase of the $EPSC_1$ amplitude by $546.3 \pm 229.8\%$ ($368.7 \pm 78.5$ pA in control conditions versus $1108.0 \pm 218.3$ pA after $HFS_{100}$; 11 pairs; $P = 0.006$; Fig. 2d–f). The magnitude of PTP was comparable ($P = 0.186$) to the condition where postsynaptic APs were omitted by voltage clamping of the postsynaptic CA3 cell with a 10 mM EGTA-containing intracellular solution (Fig. 2g). These results suggest that postsynaptic activity of the CA3 pyramidal neuron triggers an inflow of $Ca^{2+}$, putting a brake on PTP.

Previous work demonstrated that PTP at hippocampal mossy fiber synapses is mediated by changes in size of the RRP[10]. To analyze the mechanisms underlying modulation of PTP, we performed pool analysis before and after pre-postsynaptic $HFS_{100}$ in 0.1 mM and 10 mM EGTA (Supplementary Fig. 1). Whereas the RRP remained constant in the presence of 0.1 mM EGTA, it significantly increased in 10 mM EGTA after pre-postsynaptic $HFS_{100}$ (Supplementary Fig. 1). To further analyze whether changes in evoked release were occluded by spontaneous release, we examined spontaneous EPSC frequency before and after $HFS_{100}$ in 0.1 mM and 10 mM EGTA. However, spontaneous EPSC frequency was only minimally changed ($12.3 \pm 2.0$ Hz in control versus $13.4 \pm 2.0$ Hz after $HFS_{100}$ in 0.1 mM EGTA, $P = 0.10$; $11.7 \pm 2.1$ Hz in control versus $12.8 \pm 1.9$ Hz after $HFS_{100}$ in 10 mM EGTA; $P = 0.15$). Taken together, these results suggest that both PTP and its modulation converge on changes in pool size. Finally, we tested whether modulation of PTP was observed at near-physiological temperature. At ~32 °C, PTP after pre-postsynaptic $HFS_{100}$ was significantly smaller in 0.1 mM than in 10 mM EGTA ($270.4 \pm 81.6\%$ versus $543.0 \pm 167.5\%$; $P = 0.028$; Supplementary Fig. 2), corroborating our conclusions.

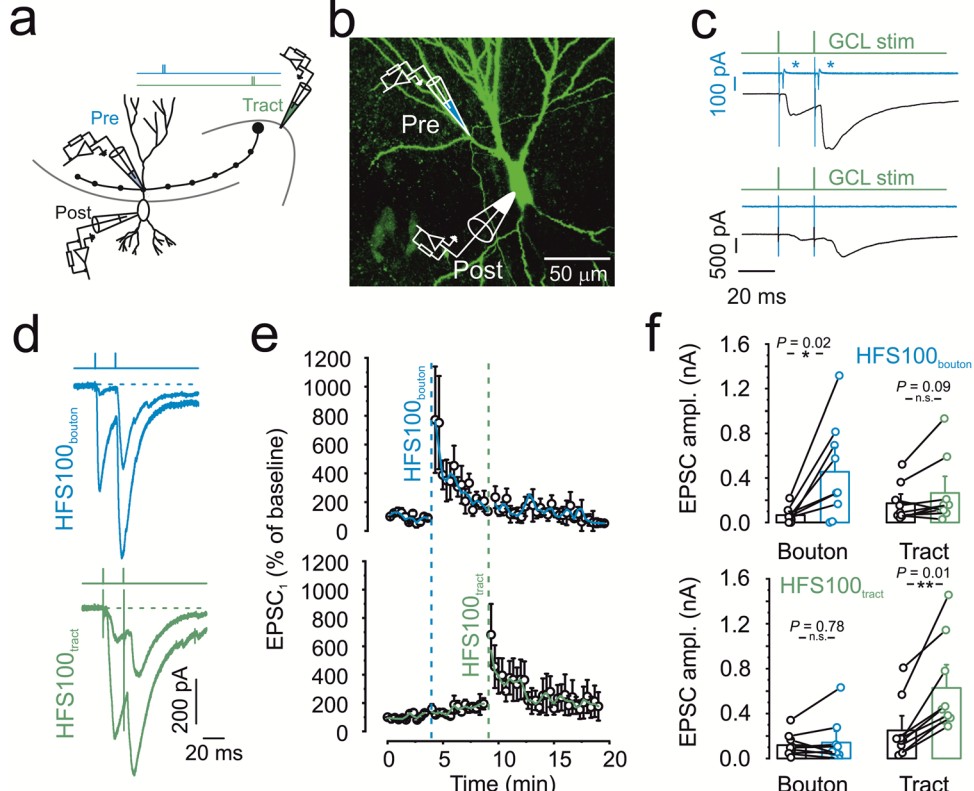

**Fig. 1 Specificity and non-cooperativity of hippocampal mossy fiber PTP. a** Schematic illustration of experimental setup. EPSCs were evoked by alternating stimulation of presynaptic terminals in the minimally invasive tight-seal bouton-attached configuration (blue pipette near proximal apical dendrite) or by tract stimulation (green pipette near granule cell layer). Two pulses were applied with 20 ms interval. **b** Maximum stack projection of confocal images of a biocytin-filled pair, stained with Alexa 488-conjugated streptavidin. Pipette on top indicates the approximate location of the tight-seal cell-attached mossy fiber bouton (MFB) recording, the pipette at the bottom shows whole-cell CA3 pyramidal cell recording. Representative micrograph; similar results were obtained in 14 biocytin-labeled pairs from a different set of experiments. **c** Selecting parallel and independent mossy fiber inputs converging on a single CA3 pyramidal cell. Top, example in which presynaptic action current analysis indicates input overlap. Green trace indicates tract stimulation, blue trace represent currents measured in tight-seal configuration at the level of the MFB, and black trace shows EPSCs. The occurrence of action currents at the MFB (indicated by blue asterisks, top panel) indicates stimulation overlap and led to the exclusion of the recordings. Bottom, example in which presynaptic action current analysis indicates input separation. **d** Representative traces showing EPSCs evoked by bouton stimulation in tight-seal cell-attached mode (top, blue traces) or by mossy fiber tract stimulation (bottom, green traces). **e** Normalized $EPSC_1$ amplitude plotted against experimental time. Blue vertical dashed line indicates delivery of a 1-s $HFS_{100}$ at the level of the MFB, held in tight-seal cell-attached mode. Green vertical dashed line indicates delivery of $HFS_{100}$ at the level of the mossy fiber tract. Top graph shows unitary EPSCs evoked by direct MFB stimulation, blue line shows running average from 2 points. Bottom graph shows compound EPSCs evoked by stimulation of the mossy fiber tract, green line shows running average. **f** Summary bar graphs showing the EPSC amplitudes derived from direct bouton stimulation (left, "Bouton") and from mossy fiber tract stimulation (right, "Tract"). Effect of bouton (top panel) or tract (bottom panel) $HFS_{100}$ as indicated. Boxes represent mean values, and circles show individual measurements. A paired non-parametric two-sided Wilcoxon signed rank test was used to test for statistical significance. * indicates $P < 0.05$, ** represents $P < 0.01$, and n.s. denotes non-significant difference ($P \geq 0.05$). Data in **e** and **f** are from 9 pairs. Error bars indicate SEM.

To further test the associativity of PTP with a more physiological induction paradigm, we performed a series of experiments in which we allowed the postsynaptic CA3 cell to spike freely (Fig. 2g). The number of postsynaptic APs observed during presynaptic $HFS_{100}$ was $8.7 \pm 1.5$ on average (ranging between 4 and 14), and also in this setting PTP ($174.2 \pm 23.7\%$; $292.8 \pm 76.5$ pA in control versus $506.2 \pm 120$ pA after $HFS_{100}$; 12 pairs) was significantly reduced as compared to control experiments performed in previous work with postsynaptic voltage-clamp ($432.5 \pm 73.9\%$; $173.7 \pm 37.8$ pA in control and $612.9 \pm 87.9$ pA after $HFS_{100}$; 12 pairs; $P = 0.0002$; Fig. 2g)[10]. The magnitude of PTP was slightly larger than that in the presynaptic $HFS_{100}$ + postsynaptic $HFS_{100}$ experiments, but the difference did not reach statistical significance ($0.1$ mM $EGTA_{post}$; $P = 0.067$; Fig. 2g). Thus, few postsynaptic spikes, not locked in time to presynaptic spikes, were sufficient to inhibit PTP. To further characterize the effects of postsynaptic firing on PTP, we plotted the magnitude of

PTP against the number of APs and the coefficient of variation of interspike intervals (ISIs). Given that PTP in control conditions affects the first three EPSCs (Fig. 2c, f), we used the average PTP value ($PTP_{1-3}$) for quantification. Correlation analysis revealed that the amount of $PTP_{1-3}$ was significantly correlated with AP number ($\rho = -0.45$; $P = 0.023$), and significantly correlated with the coefficient of variation (CV) of the ISI ($\rho = -0.64$; $P = 0.025$). Thus, patterned forms of postsynaptic activity (i.e., burst firing) may boost postsynaptic $Ca^{2+}$ entry that can generate anti-associativity of PTP. Given the correlation between the number of postsynaptic spikes and the block of PTP, we subsequently measured the magnitude of PTP in the absence of spikes, when the voltage in the postsynaptic cell was strictly controlled. We therefore loaded the postsynaptic CA3 pyramidal cell with $0.1$ mM EGTA, kept the neuron in voltage-clamp mode, and tried to trigger PTP with our standard induction protocol of 100 APs delivered presynaptically at 100 Hz (no pairing). PTP induced in

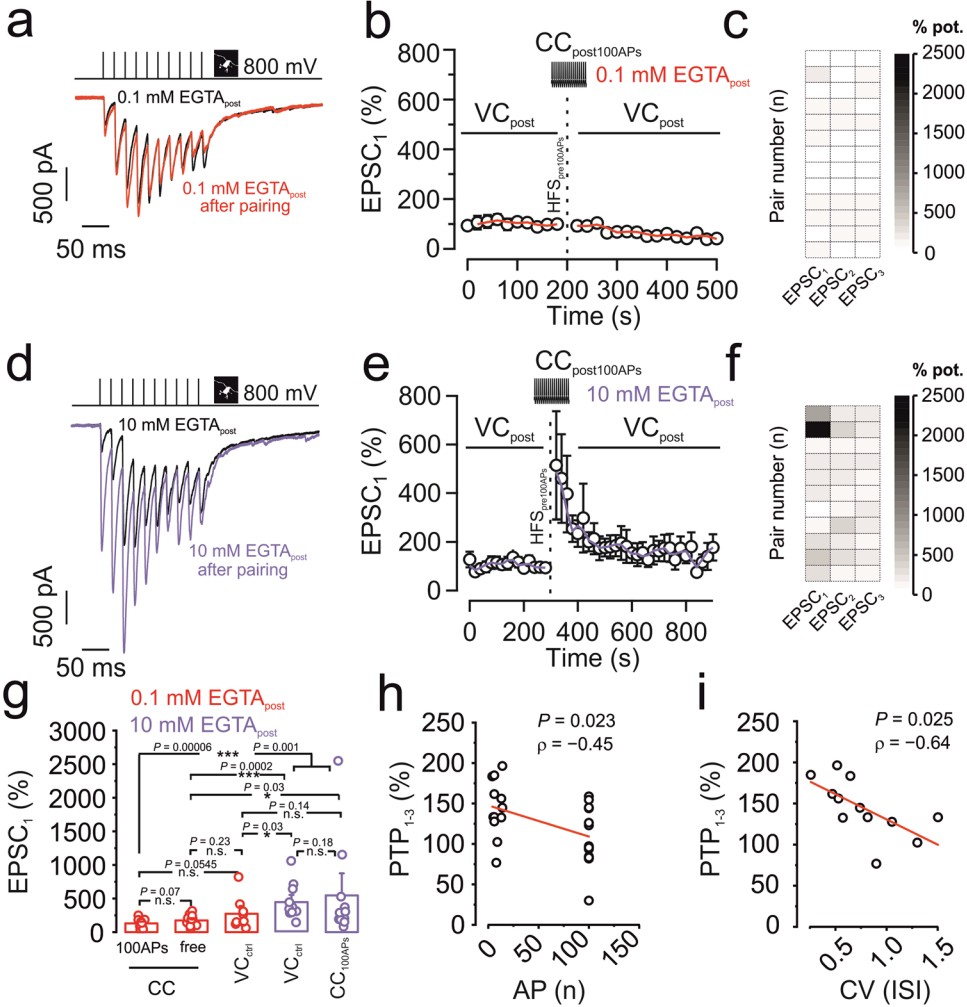

**Fig. 2 Anti-associative induction properties of mossy fiber PTP. a–c** Pre + post $HFS_{100}$ fails to induce PTP with 0.1 mM EGTA in the postsynaptic pipette.
**a** Representative traces of control (black) and after $HFS_{100}$ pairing (red). **b** Normalized $EPSC_1$ amplitude plotted versus experimental time. Black vertical dashed line indicates delivery of an $HFS_{100}$ stimulation (1 s, 100 Hz) in tight-seal cell-attached mode at the level of the MFB, combined with pairing 100 postsynaptic APs in current-clamp mode. Red line shows running average. Data from 13 pairs. **c** Intensity map showing the degree of potentiation of the first three EPSCs (in %, with control values plotted as 100%) of each single pair between an MFB and a CA3 pyramidal cell, upon pairing with 0.1 mM EGTA in the postsynaptic pipette. **d–f** Pre + post $HFS_{100}$ triggers full PTP with 10 mM EGTA in the postsynaptic pipette. **d** Representative traces of control (black) and after $HFS_{100}$ pairing (violet). **e** Normalized $EPSC_1$ amplitude plotted versus experimental time. Black vertical dashed line indicates delivery of an $HFS_{100}$ stimulation (1 s, 100 Hz) in tight-seal cell-attached mode at the level of the MFB, combined with pairing 100 postsynaptic APs in current-clamp mode. Violet line shows running average. Data from 11 pairs. **f** Intensity map showing the degree of potentiation of the first three EPSCs (in %) of each MFB-CA3 pyramidal neuron pair, with 10 mM EGTA postsynaptically. **g** Summary bar graphs showing the percentage of potentiation of $EPSC_1$ by pairing 100 presynaptic APs with or without postsynaptic APs with different concentrations of the $Ca^{2+}$ chelator EGTA. Red bar graphs represent pairs where 0.1 mM EGTA was included in the postsynaptic pipette. Violet bar graphs represent pairs where 10 mM EGTA was included in the postsynaptic pipette. $CC_{100APs}$ refers to **a–c** and **d–f** (above). $CC_{free}$ refers to experiments where the postsynaptic cell was allowed to spike freely in current-clamp mode upon presynaptic $HFS_{100}$. $VC_{ctrl}$ in the presence of 10 mM EGTA refers to data published in Vandael et al.[10], where the postsynaptic cell was under voltage-clamp control to prevent spiking. Boxes represent mean values, and circles show individual measurements. A non-paired non-parametric two-sided Mann–Whitney U test was used to test for statistical significance. * indicates $P < 0.05$, ** $P < 0.01$, *** $P < 0.001$, and n.s. denotes non-significant difference ($P \geq 0.05$). $P$ values are given without multiple comparison correction. Data from 13, 12, 9, 12, and 11 pairs. **h** Scatter plot of magnitude of PTP against number of APs of the postsynaptic CA3 pyramidal neuron. The degree of PTP refers to the average PTP obtained for the first three EPSCs ($EPSC_{1–3}$). Note a significant correlation. Line represents linear regression to the data points (Pearson's correlation coefficient $\rho = -0.45$; $P = 0.023$; 25 pairs; 13 with $CC_{100APs}$ and 12 with $CC_{free}$). **i** Scatter plot of magnitude of PTP against coefficient of variation (CV) of ISI duration of the postsynaptic CA3 pyramidal neuron. The degree of PTP refers to the average PTP obtained for the first three EPSCs ($EPSC_{1–3}$). Data points taken from postsynaptic $CC_{free}$ recordings upon presynaptic $HFS_{100}$. Note a significant correlation. Line represents linear regression to the data points (Pearson's correlation coefficient $\rho = -0.64$; $P = 0.025$; 12 pairs). Error bars indicate SEM.

voltage-clamp conditions with 0.1 mM EGTA was intermediate (273.0 ± 82.2%; from 169.7 ± 56.9 pA to 290.0 ± 72.4 pA; 9 pairs) between PTP with 100 APs in current clamp in 0.1 mM EGTA ($P = 0.055$; Fig. 2g) and PTP with voltage-clamp or current-clamp induction with 100 APs in 10 mM EGTA, although all differences

were near the limits of statistical significance ($P = 0.03$ and $P = 0.144$, respectively; Fig. 2g). Taken together, our results suggest that patterned forms of postsynaptic activity are sufficient to put a brake on mossy fiber PTP. Recent work demonstrated that PTP could be attributed to a transient increase in the size of the RRP[10].

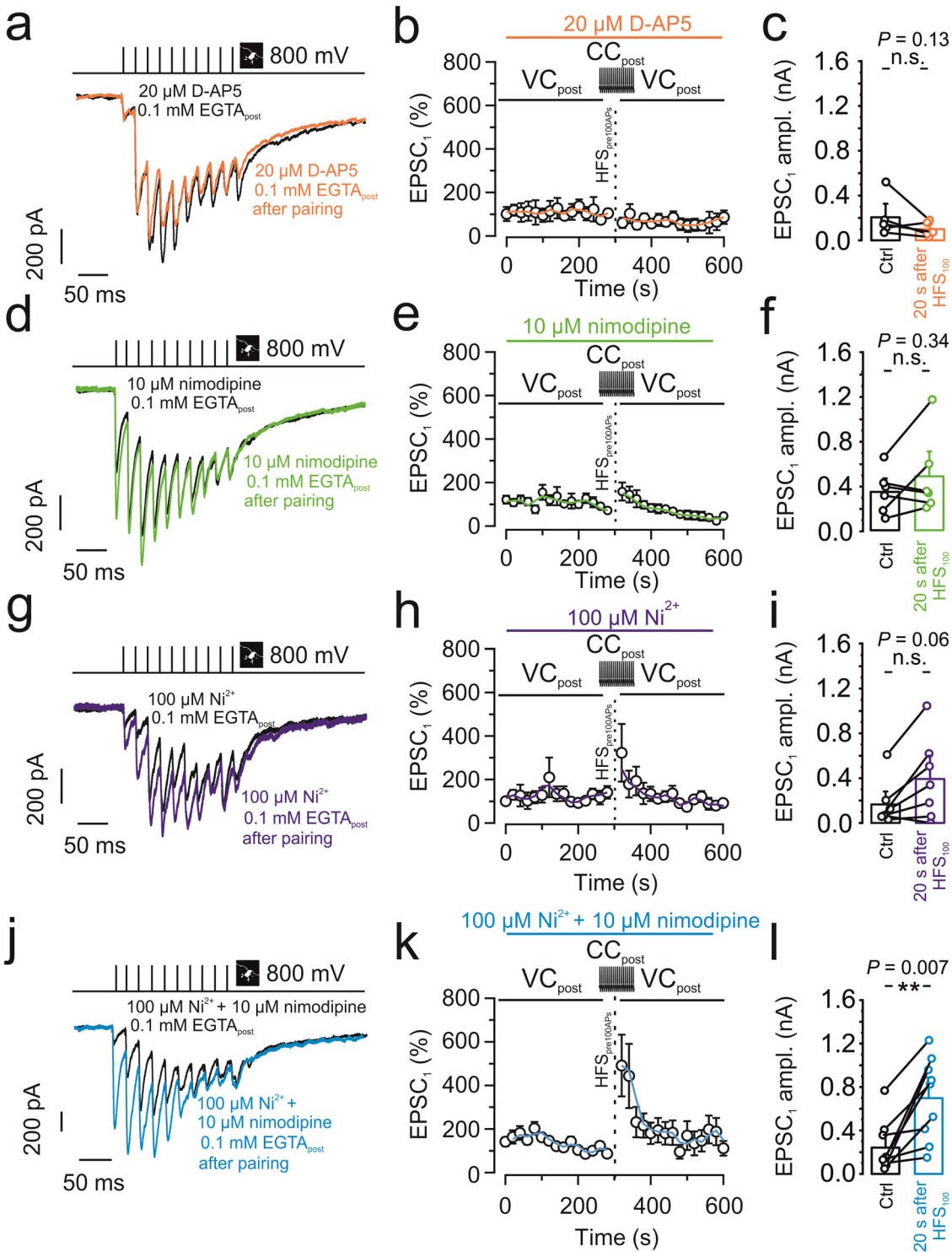

Thus, we asked whether changes in PTP were also generated by a vesicle pool mechanism, and plotted relative changes in RRP against relative changes in PTP. The two measures were highly correlated ($\rho = 0.993$; $P < 0.001$; Supplementary Fig. 1), suggesting that regulation of PTP predominantly affects the size of the RRP.

**Mechanisms underlying anti-associative PTP.** To elucidate the signaling mechanisms underlying anti-associative PTP induction, we dissected the involvement of pre- and postsynaptic signaling

cascades. We first tried to determine the source for the rise in postsynaptic $Ca^{2+}$ (Fig. 3). As N-methyl-D-aspartate (NMDA)-type glutamate receptors are coincidence detectors, and are also present at mossy fiber–CA3 synapses[2,17,18], we first tested the effects of NMDA receptor blockers. We obtained a recording with 0.1 mM EGTA in the postsynaptic pipette, bath-applied 20 µM D-AP5, and after obtaining a baseline of 5 min applied an induction protocol pairing 100 presynaptic APs with postsynaptic APs (Fig. 3a–c). D-AP5 failed to recover PTP ($59.9 \pm 22.8\%$); $EPSC_1$ amplitude was $205.7 \pm 90.4$ pA in control conditions and dropped to $99.6 \pm 32.9$ pA after $HFS_{100}$ ($P = 0.13$; 5 pairs; Fig. 3a–c). We

**Fig. 3 Anti-associative PTP involves postsynaptic Ca$^{2+}$ inflow deriving from R- and L-type Ca$^{2+}$ channels. a–c** 20 μM D-AP5 did not restore PTP after pairing 100 presynaptic APs with postsynaptic APs with 0.1 mM EGTA in the postsynaptic pipette. **a** Representative traces of control (black) and after PTP induction (orange). **b** Normalized EPSC$_1$ amplitude plotted against experimental time. Black vertical dashed line indicates PTP induction, pairing HFS$_{100}$ stimulation (1 s, 100 Hz) in the presynaptic terminal (tight-seal cell-attached mode) with spiking in the postsynaptic neuron (whole-cell current-clamp configuration). Orange line shows running average. Orange horizontal bar on top indicates the application period of the NMDA receptor blocker D-AP5. **c** Summary bar graph showing the EPSC$_1$ amplitude before ("Ctrl"; black) and after pairing 100 presynaptic APs with postsynaptic APs ("20 s after HFS$_{100}$"; orange) in the presence of 20 μM D-AP5. **d–f** Similar experiments as shown in **a–c**, but with 10 μM of the L-type Ca$^{2+}$ channel blocker nimodipine (green). Nimodipine alone did not restore PTP. **g–i** Similar experiments as shown in **a–c**, but with 100 μM of the T-type and R-type Ca$^{2+}$ channel blocker Ni$^{2+}$ (purple). Ni$^{2+}$ led to a trend towards recovery of PTP. **j–l** Similar experiments as shown in **a–c**, but with 10 μM of the L-type Ca$^{2+}$ channel blocker nimodipine and 100 μM of the T- or R-type Ca$^{2+}$ channel blocker Ni$^{2+}$ (blue). Combined application of the Ca$^{2+}$ channel blockers largely restored PTP. **c, f, i, l** Boxes represent mean values, and circles show individual measurements. A paired non-parametric two-sided Wilcoxon signed rank test was used to test for statistical significance. ** indicates $P < 0.01$, and n.s. denotes non-significant difference ($P \geq 0.05$). Data in (**b, c**), (**e, f**), (**h, i**), and (**k, l**) are from 5, 6, 7, and 9 pairs, respectively. Error bars indicate SEM.

therefore tested alternative Ca$^{2+}$ sources. Application of 10 μM of the L-type Ca$^{2+}$ channel blocker nimodipine only slightly increased (158.9 ± 44%) the amount of PTP induced by pairing of pre- and postsynaptic APs (EPSC$_1$: 343.8 ± 87.4 pA in control versus 482.9 ± 162.3 pA after HFS$_{100}$; $P = 0.34$; 6 pairs; Fig. 3d–f). Thus, L-type Ca$^{2+}$ channels contributed only slightly to the relevant postsynaptic Ca$^{2+}$ inflow. Next, we examined the possible contribution of T-type and R-type Ca$^{2+}$ channels by applying Ni$^{2+}$ at 100 μM[19]. Application of 100 μM Ni$^{2+}$ led to a trend towards recovery of PTP induced by pre-postsynaptic AP pairing (305.4 ± 145.7%; 166.9 ± 83.2 pA in control versus 383.36 ± 152.6 pA after HFS$_{100}$; $P = 0.06$; 7 pairs; Fig. 3g–i). Consistent with previous results[20,21], 100 μM Ni$^{2+}$ did not affect basal synaptic transmission. Finally, we tested the combined effects of 10 μM nimodipine and 100 μM Ni$^{2+}$. Combined application of both Ca$^{2+}$ channel blockers led to a full recovery of PTP induced by pre-postsynaptic AP pairing (491.2 ± 150.2%; 242.4 ± 82.1 pA in control versus 697.8 ± 133.4 pA after HFS$_{100}$; $P = 0.007$; 9 pairs; Fig. 3j–l). In parallel to the rescue of PTP in terms of EPSC amplitude, Ca$^{2+}$ channel blockers rescued potentiation of pool size (Supplementary Fig. 3). Given that T-type channels show voltage-dependent inactivation around the resting potential of CA3 pyramidal neurons ($-65$ mV)[22], our results are most consistent with the hypothesis that postsynaptic R- and L-type Ca$^{2+}$ channels are synergistically responsible for the apparently transsynaptic modulation of mossy fiber PTP.

Finally, we examined the molecular targets at the presynaptic terminal responsible for the transsynaptic regulation of PTP (Fig. 4). At immature MFB-CA3 pyramidal neuron synapses, postsynaptic spiking is known to lead to endocannabinoid release and cannabinoid receptor 1 (CB1) activation at the presynaptic terminal[23]. This results in suppression of neurotransmitter release and could in theory explain our findings. However, 5 μM of the CB1 receptor antagonist / inverse agonist AM251 did not significantly rescue PTP induced by pre-postsynaptic AP pairing (154.8 ± 67%; 303.4 ± 130.8 pA in control conditions versus 422.8 ± 217.9 pA after HFS$_{100}$; $P = 0.25$; 6 pairs; Fig. 4a–c). Another possible mechanism would be glutamate-mediated inhibition via the presynaptic metabotropic glutamate receptor 2 (mGluR2), which is highly enriched in hippocampal mossy fiber axons and terminals[24–26]. Notably, application of 3 μM of the group II (mGluR2 / mGluR3) antagonist LY341495 largely rescued PTP induced by pre-postsynaptic AP pairing (343.6 ± 98.5%; 273.9 ± 75.9 pA in control versus 805.1 ± 239.3 pA after HFS$_{100}$; $P = 0.005$; 10 pairs; Fig. 4d–f). In parallel to the rescue of PTP of EPSC amplitude, LY341495 rescued potentiation of pool size (Supplementary Fig. 3).

Given that the anti-associative mechanism was dependent on postsynaptic Ca$^{2+}$ chelators, we tested the contribution of retrograde vesicular glutamate signaling. Since glutamate transport into vesicles typically requires a proton gradient, we reasoned that interfering with this mechanism might mimic the effect of the mGluR2 / mGluR3 antagonist LY341495. To specifically target the postsynaptic neuron, we loaded CA3 pyramidal neurons with the v-type H$^+$-ATPase inhibitor concanamycin A (20–40 nM) via the patch pipette. Postsynaptic specificity was demonstrated by the lack of inhibition of presynaptic mediated neurotransmitter release (baseline, Fig. 4h). Interestingly, treatment with concanamycin A led to a moderate but highly significant recovery of PTP induced by pre-postsynaptic AP pairing (267.5 ± 40.5%; 246.8 ± 62.5 pA in control versus 554.2 ± 112.3 pA after HFS$_{100}$; $P = 0.003$; 12 pairs; Fig. 4g–i).

Comparison of PTP after various pharmacological manipulations with PTP in 0.1 mM and 10 mM EGTA suggested that Ca$^{2+}$ channel blockers, LY341495, and concanamycin A recovered PTP to a large extent (Supplementary Fig. 4). Taken together, our results may suggest a retrograde signaling mechanism, in which postsynaptic activity induces an increase in postsynaptic Ca$^{2+}$ concentration, vesicular release of a retrograde messenger, probably glutamate, from the dendrites of the postsynaptic cell[27,28], and subsequent activation of presynaptic mGluRs (Fig. 5).

## Discussion

Synapse specificity, cooperativity, and associativity are hallmark features of synaptic plasticity at glutamatergic synapses. However, rigorous testing of these properties is often difficult, because it requires defined stimulation of individual synaptic inputs. To determine specificity, cooperativity, and associativity of hippocampal mossy fiber PTP, we combined paired recording between mossy fiber terminals and postsynaptic CA3 pyramidal neurons in rat brain slices[9,10] with extracellular tract stimulation. This allowed us to stimulate single inputs, as required for cooperativity analysis, to validate the non-overlapping nature of the inputs by presynaptic action current measurements, essential for specificity experiments, and to precisely control pre- and postsynaptic activity, critically important for characterization of associativity.

Using this tightly controlled approach, we demonstrate that hippocampal mossy fiber PTP shows synapse specificity, but lacks cooperativity and associativity. PTP induction is not only non-associative, but rather shows anti-associative properties. Non-cooperativity might have been expected, given that PTP can be robustly induced by single bouton stimulation[9,10]. In contrast, the anti-associativity was surprising. Although previous work showed that mossy fiber synaptic plasticity can be enhanced by postsynaptic hyperpolarization[29], this is, to the best of our knowledge, the first description of anti-associative properties of short-term plasticity at a glutamatergic synapse. The unique induction rules of PTP provide the synapse with complete autonomy and a

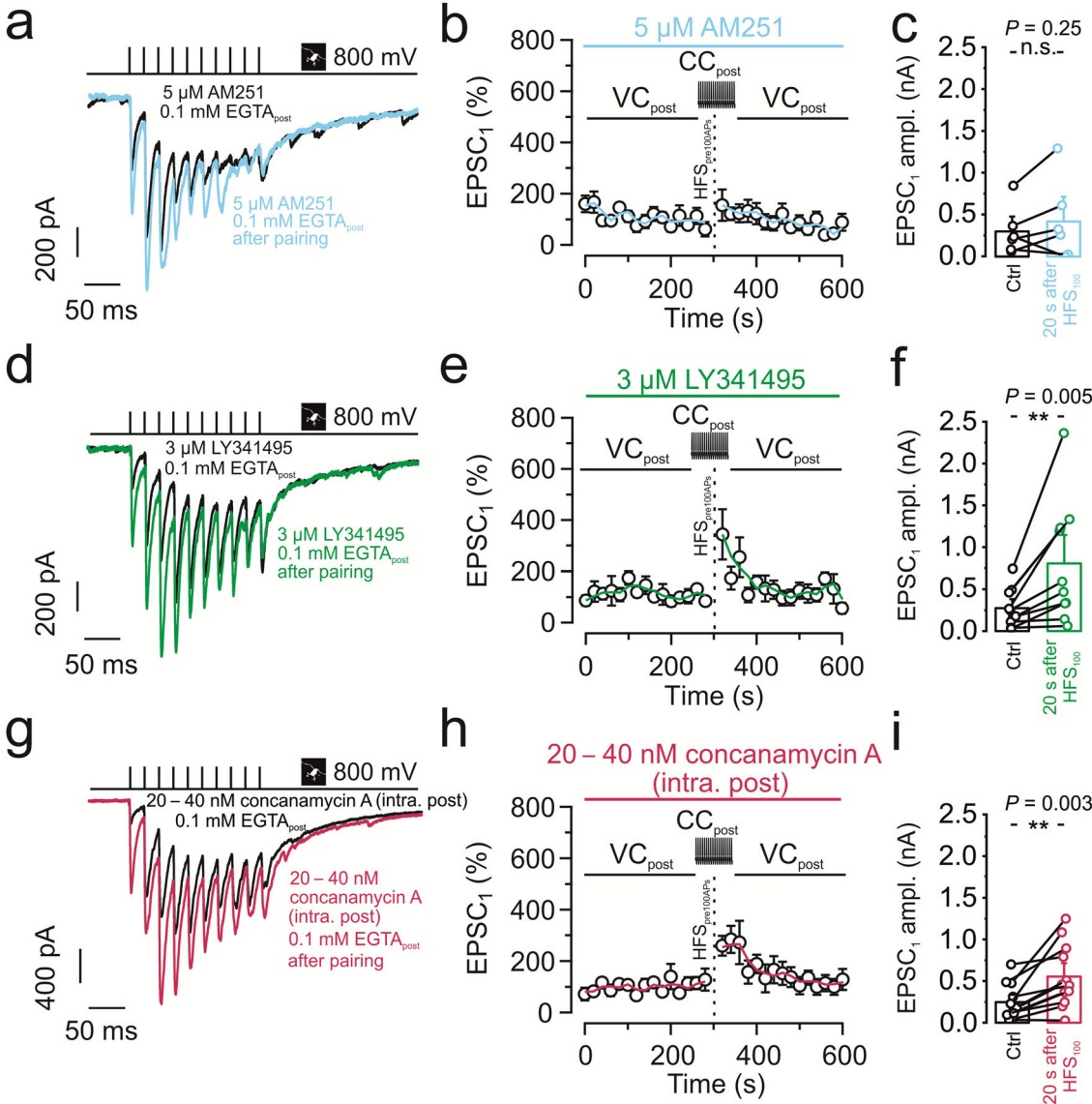

**Fig. 4 Anti-associative PTP involves retrograde glutamate signaling and activation of presynaptic mGluR2/3 receptors. a–c** 5 μM of the CB1 receptor antagonist / inverse agonist AM251 did not restore PTP after pairing 100 presynaptic APs with postsynaptic APs with 0.1 mM EGTA in the postsynaptic pipette. **a** Representative traces of control (black) and after PTP induction (light blue). **b** Normalized $EPSC_1$ amplitude plotted versus experimental time. Black vertical dashed line indicates PTP induction, pairing $HFS_{100}$ stimulation (1 s, 100 Hz) in the presynaptic terminal (tight-seal cell-attached mode) with spiking in the postsynaptic neuron (whole-cell current-clamp configuration). Light blue line shows running average. Light blue horizontal bar on top indicates the application period of the CB1 receptor antagonist / inverse agonist. **c** Summary bar graph showing the $EPSC_1$ amplitude before ("Ctrl"; black) and after pairing 100 presynaptic APs with postsynaptic APs ("20 s after $HFS_{100}$"; light blue) in the presence of 5 μM AM251. **d–f** 3 μM of the mGluR2 / mGluR3 antagonist LY341495 largely restored PTP after pairing 100 presynaptic APs with postsynaptic APs with 0.1 mM EGTA in the postsynaptic pipette. **d** Representative traces of control (black) and after PTP induction (green). **e** Normalized $EPSC_1$ amplitude plotted versus experimental time. Black vertical dashed line indicates PTP induction, pairing $HFS_{100}$ stimulation (1 s, 100 Hz) in the presynaptic terminal (tight-seal cell-attached mode) with spiking in the postsynaptic neuron (whole-cell current-clamp configuration). Green line shows running average. Green horizontal bar on top indicates the application period of the mGluR2/3 receptor antagonist. **f** Summary bar graph showing the $EPSC_1$ amplitude before ("Ctrl"; black) and after pairing 100 presynaptic APs with postsynaptic APs ("20 s after $HFS_{100}$"; green) in the presence of 3 μM LY341495. **g–i** 20–40 nM of the $H^+$-ATPase inhibitor concanamycin A (also called folimycin) in the postsynaptic pipette largely restored PTP after pairing 100 presynaptic APs with postsynaptic APs. **g** Representative traces of control (black) and after PTP induction (magenta). **h** Normalized $EPSC_1$ amplitude plotted versus experimental time. Black vertical dashed line indicates PTP induction, pairing $HFS_{100}$ stimulation (1 s, 100 Hz) in the presynaptic terminal (tight-seal cell-attached mode) with spiking in the postsynaptic neuron (whole-cell current-clamp configuration). Magenta line shows running average. Magenta horizontal bar on top indicates the application period of the $H^+$-ATPase inhibitor. Note that the time course of PTP in the presence of concanamycin A is slightly different from that of control data sets, suggesting that a remaining suppression is more short-lasting. **i** Summary bar graph showing the $EPSC_1$ amplitude before ("Ctrl"; black) and after pairing 100 presynaptic APs with postsynaptic APs ("20 s after $HFS_{100}$"; magenta) in the presence of 20–40 nM of concanamycin A. **c, f, i** Boxes represent mean values, and circles show individual measurements. A paired non-parametric two-sided Wilcoxon signed rank test was used to test for statistical significance. ** indicates $P < 0.01$, and n.s. denotes non-significant difference ($P \geq 0.05$). Data in (**b, c**), (**e, f**), and (**h, i**) are from 6, 10, and 12 pairs, respectively. Error bars indicate SEM.

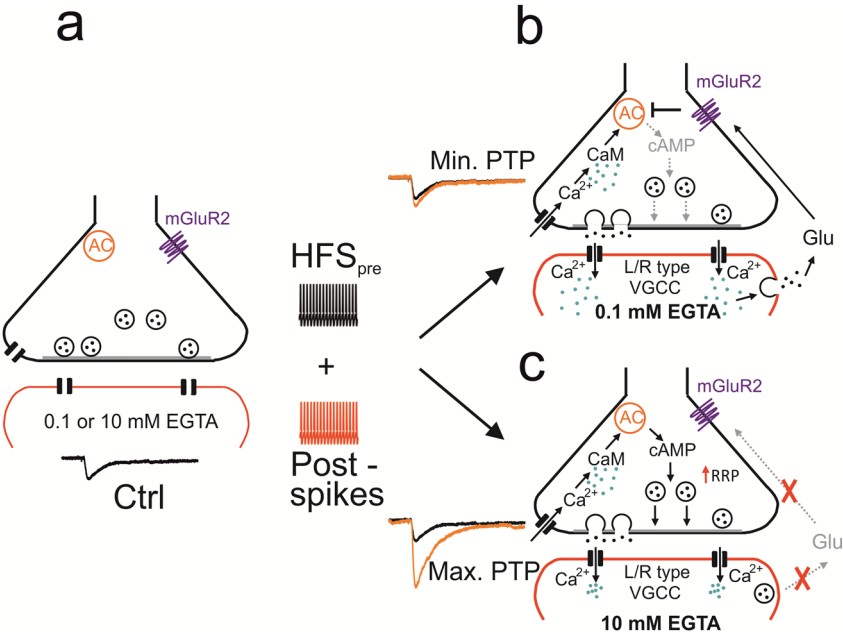

**Fig. 5 Mechanisms of PTP and possible signaling pathways underlying transsynaptic modulation at the MFB–CA3 pyramidal neuron synapse.**
**a** MFB–CA3 synapse during resting conditions. Presynaptic terminal in black on top, postsynaptic compartment in red at the bottom. AC stands for adenylyl cyclase, mGluR2 indicates extrasynaptic metabotropic glutamate receptor 2 (and mGluR3 expressed at lower levels). **b** MFB–CA3 synapse after tetanic stimulation, leading to postsynaptic activity, in the presence of low (0.1 mM) concentrations of the $Ca^{2+}$ chelator EGTA in the postsynaptic cell. CaM stands for calmodulin, VGCC for voltage-gated $Ca^{2+}$ channels, and Glu for glutamate. $Ca^{2+}$ ions indicated as green dots, glutamate molecules as black dots. Note that in this scenario tetanic stimulation does not trigger PTP, due to retrograde vesicular glutamate signaling. **c** Same as for **b** but in the presence of high concentration (10 mM) of the $Ca^{2+}$ chelator EGTA in the postsynaptic cell. Note that in this scenario tetanic stimulation does trigger PTP, due to the lack of retrograde vesicular glutamate signaling. Transsynaptic modulation of PTP induction may not only occur with the exogenous $Ca^{2+}$ chelator EGTA, but also with endogenous $Ca^{2+}$ buffers expressed in the postsynaptic neurons. This may convey both anti-associativity and target cell-specificity to the PTP induction mechanism[51].

powerful feedback mechanism to prevent excessive detonation in the hippocampal network. Future work will reveal the synapse-specificity of suppression of PTP.

Our results show that anti-associativity can be observed under conditions in which postsynaptic $HFS_{100}$ is induced by post-synaptic current injections and conditions in which the postsynaptic CA3 pyramidal neuron freely fires in response to mossy fiber terminal stimulation during PTP induction. This corroborates the physiological significance of the observed anti-associativity mechanism. Interestingly, the suppressive effect on PTP is correlated with both the number of spikes and the coefficient of variation of the spike interval. This suggests that the anti-associativity is facilitated by burst firing in the postsynaptic neuron. This is consistent with the involvement of R-type $Ca^{2+}$ channels, which have been reported to promote burst firing in subicular neurons[30].

What are the mechanisms underlying anti-associative PTP induction? Our results show that both $Ca^{2+}$ chelators introduced into the postsynaptic cell and L- and R-type $Ca^{2+}$ channel blockers, when applied in combination, rescued PTP. Thus, $Ca^{2+}$ inflow through postsynaptic voltage-gated $Ca^{2+}$ channels plays a critical role in the suppression of PTP. This is consistent with the presence of R-type channel immunoreactivity in thorny excrescences of hippocampal mossy fiber synapses[31] and high-voltage activated, $Ni^{2+}$-sensitive $Ca^{2+}$ channels (putatively R-type) in the dendritic shafts of hippocampal pyramidal neurons[32]. It is interesting that voltage-clamp does not completely mimic the effects of 10 mM EGTA. This may suggest that R-type $Ca^{2+}$ channels are activated by local dendritic depolarizations, partially escaping voltage-clamp control[33]. Earlier work demonstrated that mossy fiber PTP is generated by an increase in the size of the

readily releasable synaptic vesicle pool[10]. As these mechanisms operate in the presynaptic terminals, retrograde transsynaptic signaling mechanisms must be involved[34]. Our results identify mGluR2 or mGluR3, probably activated by dendritically released glutamate, as the critical link between postsynaptic spiking activity and presynaptic terminal function. mGluR2 and mGluR3 are known to couple to $G_i$ and thereby inhibit adenylyl cyclase[24,35]. Adenylyl cyclase, in turn, regulates the size of the RRP in mossy fiber terminals[10]. Thus, postsynaptic activity could regulate presynaptic vesicle pool size (Fig. 5). However, we cannot rule out that changes in release probability contribute to these effects. mGluR2 and mGluR3 have been reported to decrease presynaptic $Ca^{2+}$ inflow, which will suppress release probability[36], although direct effects on the release machinery also have been suggested[37].

How exactly glutamate is released from postsynaptic CA3 cells remains unclear. Our finding that the vacuolar-type $H^+$-ATPase inhibitor concanamycin A introduced in the postsynaptic pipette reduced the suppression of PTP may be consistent with a vesicular release mechanism, in which glutamate accumulation depends on established proton gradients. Consistent with this hypothesis, thorny excrescences of CA3 pyramidal neurons show a pronounced spine apparatus and are enriched in multivesicular bodies[38]. More work is needed to disentangle the role of these structures in retrograde signaling. If dendritically released glutamate inhibits PTP via mGluR2 or 3, why does glutamate released from presynaptic terminals not have the same effect? mGluR2 or 3 show a relatively low affinity for glutamate[35]. Furthermore, mGluR2 is primarily located in extrasynaptic regions of mossy fiber axons[26]. In combination, receptor properties and subcellular localization may facilitate activation of the pathway by dendritic release.

Previous work identified several mechanisms that enhance synaptic efficacy at hippocampal mossy fiber synapses, including facilitation, post-tetanic potentiation, and long-term potentiation[3,8,9,39]. This led to the suggestion that the hippocampal mossy fiber–CA3 pyramidal neuron synapse can operate as a conditional or plasticity-dependent full detonator[9,40–42]. However, the abundance of multiple forms of potentiating plasticity raises the question of how these detonation mechanisms might be controlled. Transsynaptic modulation of PTP induction may represent a simple, but powerful mechanism to prevent excessive potentiation in the hippocampal network, and to implement a brake on mossy fiber detonation. This may be important to keep the balance between excitation and inhibition in the network.

Previous analysis in which PTP was probed with field-potential recordings indicated that potentiation is highly robust[3,43]. Given the detonator properties of mossy fiber synapses, postsynaptic spiking is likely to be induced under these conditions. How is it possible that PTP can be reliably induced? Our results suggest that both postsynaptic spiking and postsynaptic $Ca^{2+}$ buffering regulates the magnitude of PTP (Fig. 2g). One possibility is that the postsynaptic $Ca^{2+}$ buffer capacity is higher than that of 0.1 mM EGTA. More work is needed to test this hypothesis.

The present findings may have major implications for the memory function of the hippocampal network. A prevalent model of hippocampal memory suggests that the mossy fiber synapse acts as a "teacher" that triggers the storage and recall of memories in the downstream CA3 region[44,45]. In this model, the mossy fiber synapse is often viewed as a conditional detonator, in which the efficacy of the synapse depends on prior presynaptic activity[9,41]. Our results challenge this view, showing that the detonation properties of the synapse depend on both pre- and postsynaptic activity in a complex manner. This might be a powerful mechanism to ensure that storage and recall are separated and that new information is preferentially stored in silent, non-coding CA3 pyramidal neurons[46]. Thus, the mossy fiber synapse may act as a "smart teacher", which may help to maximize the storage capacity in the system. Future work combining electrophysiological analysis, network modeling, and behavioral analysis will be needed to further test this hypothesis.

## Methods

**Animal experiments.** Paired pre- and postsynaptic recordings in vitro were carried out on 17- to 23-day-old Wistar rats (RRID:RGD_13508588; weight: 55–65 g). Animals were housed under a reversed light cycle (dark: 7:00 am–7:00 pm, light: 7:00 pm–7:00 am). For experiments, both male and female animals were used. All experiments were carried out in strict accordance with institutional, national, and European guidelines for animal experimentation, and approved by the Bundesministerium für Wissenschaft, Forschung und Wirtschaft of Austria (A. Haslinger).

**Paired recordings from mossy fiber terminals.** Transverse hippocampal slices (350–400 μm thick) were prepared according to previously published protocols[10,47,48]. Animals were anesthetized using isoflurane and killed by rapid decapitation. Slices were cut from the right or left hemisphere in ice-cold, sucrose-containing extracellular solution using a vibratome (VT1200, Leica Microsystems), incubated in a maintenance chamber at ~35 °C for 30–45 min, and subsequently stored at room temperature. Cutting solution contained 64 mM NaCl, 25 mM NaHCO₃, 2.5 mM KCl, 1.25 mM NaH₂PO₄, 10 mM glucose, 120 mM sucrose, 0.5 mM CaCl₂, and 7 mM MgCl₂. Storage solution contained 87 mM NaCl, 25 mM NaHCO₃, 2.5 mM KCl, 1.25 mM NaH₂PO₄, 10 mM glucose, 75 mM sucrose, 0.5 mM CaCl₂, and 7 mM MgCl₂ (equilibrated with 95% O₂ and 5% CO₂). Experiments were performed at room temperature (24.1 ± 0.2 °C; range: 21–26 °C), and, in a subset of experiments, at near-physiological temperature (32.4 ± 0.2 °C; range: 31–34 °C). Before onset of the experiment, slices were placed in the recording chamber and superfused with artificial cerebrospinal fluid (ACSF; 125 mM NaCl, 25 mM NaHCO₃, 2.5 mM KCl, 1.25 mM NaH₂PO₄, 2 mM CaCl₂, and 1 mM MgCl₂, equilibrated with 95% O₂ and 5% CO₂) for at least 15 min before recording.

Subcellular patch-clamp recordings from MFBs and simultaneous recordings from pyramidal neurons in the CA3b and CA3c region of the hippocampus were performed under visual control provided by infrared differential interference contrast videomicroscopy[10,47,48]. Presynaptic and postsynaptic recording pipettes

were fabricated from borosilicate glass tubing (2.0 mm outer diameter, 1.0 mm inner diameter) and had open-tip resistances of 10–20 MΩ and 3–7 MΩ, respectively. For tight-seal, bouton-attached stimulation under voltage-clamp conditions, the presynaptic pipette contained a K⁺-based intracellular solution (130 mM K-gluconate, 2 mM KCl, 2 mM MgCl₂, 2 mM Na₂ATP, 10 mM HEPES, 10 mM EGTA, and 0.2% biocytin; or 140 mM KCl, 2 mM MgCl₂, 4 mM Na₂ATP, 0.3 mM Na₂GTP, 10 mM HEPES, 0.1 mM EGTA, and 0.2% biocytin; pH adjusted to 7.28 with KOH), allowing us to minimally invasively stimulate boutons in the tight-seal cell-attached configuration. In the cell-attached configuration, seal resistance was >1 GΩ and holding potential was set at –70 mV to minimize the holding current. APs in MFBs were evoked by brief voltage pulses (amplitude 800 mV, duration 0.1 ms). Mossy fiber terminals had diameters of ~2–5 μm, in agreement with the previously reported range of diameters of MFBs in light and electron microscopy studies. For experiments in all figures, trains of 10 stimuli at 50 Hz were delivered every 20 s, except Fig. 1, in which pairs of stimuli were applied, and Supplementary Fig. 2 (near-physiological temperature), in which trains of 3 or 10 stimuli were used. For tract stimulation experiments, a large pipette (tip diameter 5–10 μm) filled with 1 M NaCl was placed in the subgranular zone of the dentate gyrus. Stimulation was performed using a stimulus isolation unit; stimulation intensity was ~30 V (range: 10–100 V).

Postsynaptic recording pipettes contained an internal solution containing 130 mM K-gluconate, 2 mM or 20 mM KCl, 2 mM MgCl₂, 2 mM Na₂ATP, 10 mM HEPES, 10 mM or 0.1 mM EGTA, and 0.2% biocytin (pH adjusted to 7.28 with KOH). In a subset of recordings 0.3 mM GTP was added to a postsynaptic internal solution, and 20 mM glucose was added to adjust the osmolarity in K⁺-based intracellular solution with 2 mM KCl. Cells with initial resting potentials more positive than –50 mV were discarded. For whole-cell voltage-clamp recordings from postsynaptic CA3 pyramidal neurons, the membrane potential was set to –70 or –80 mV. Only recordings with <300 pA leakage current were included in the analyses. Postsynaptic series resistance was kept below 10 MΩ. In the voltage-clamp mode, series resistance was uncompensated, but carefully monitored with a test pulse following each data acquisition sweep. Only recordings with stable series resistance were included in the analyses (100 s time interval before HFS: 8.65 ± 0.29 MΩ; 60 s time interval after HFS: 9.09 ± 0.32 MΩ; 136 mossy fiber terminal–CA3 pyramidal neuron recordings). In experiments in which PTP was induced in postsynaptic current-clamp mode, postsynaptic CA3 pyramidal neurons were switched from voltage-clamp to current-clamp mode during the induction period (~10 s). In experiments in which the postsynaptic neuron was stimulated (CC₁₀₀ₐₚₛ), trains of current pulses were applied. Short pulses of high intensity (0.1-ms duration; 20-nA amplitude) were used to ensure reliable and temporally precise stimulation. In experiments in which the postsynaptic neuron was allowed to spike freely (CC_free), current pulses were omitted. In a fraction of recordings, a small hyperpolarizing current was injected into the postsynaptic cell to ensure the absence of spontaneous spiking (mean current: –68.1 ± 23.5 pA). The average membrane potential before HFS was –61.78 ± 0.52 mV (103 pairs).

Paired recordings (presynaptic tight-seal, bouton-attached stimulation, postsynaptic whole-cell recording) were stable for up to 30 min. Nimodipine, D-AP5, and AM251 were dissolved in dimethyl sulfoxide (DMSO) at a concentration of 10 mM and added to the ACSF. Concanamycin A was dissolved in DMSO at a concentration of 20 μM and added to the pipette solution. LY341495 (free base) was dissolved in DMSO, LY341495 (disodium salt) was dissolved in water at a concentration of 6 mM. Free base and disodium salt of LY341495 gave identical results, therefore data were pooled. The final concentration of DMSO in the ACSF was ≤0.2%. Nimodipine, D-AP5, AM251, concanamycin A, and LY341495 were from Tocris, other chemicals were from Sigma-Aldrich.

**Data analysis.** Data were acquired with a Multiclamp 700 A amplifier, low-pass filtered at 10 kHz, and digitized at 40 or 100 kHz using a CED 1401 plus or power1401 mkII interface (Cambridge Electronic Design, Cambridge, UK). Pulse generation and data acquisition were performed using FPulse version 3.3.3 (U. Fröbe, Physiological Institute, University of Freiburg, Germany). Data were analyzed using Stimfit version 0.14 or 0.15[49] and Igor Pro 6.37 (Wavemetrics). Experiments with unstable baseline were excluded from further analysis. For peak detection, the analysis time window was set between 1 and 19 ms after the stimulus. Since PTP peaked in between 20 s and 60 s after HFS₁₀₀, we compared the peak amplitude of the largest EPSC in this time interval with that of the average EPSC in a time interval 100 s before HFS, which was defined as PTP_max. For Fig. 2h, i, we used the average of PTP_max of the first three EPSCs to enhance the reliability of recordings. Pool analysis was performed by linear regression of cumulative EPSC amplitudes[10,50]. Spontaneous EPSCs were detected using a template-fit analysis[2].

**Statistics and conventions.** Statistical significance was assessed using a two-sided Wilcoxon signed rank test for paired comparisons or a two-sided Mann–Whitney U test for unpaired comparisons at the significance level (P) indicated, using Python 2. Multiple comparisons were performed with a one-sided Kruskal–Wallis test. Correlation analysis was performed based on Pearson's correlation coefficient. Values are given as mean ± standard error of the mean (SEM). Error bars in the figures also represent the SEM. For graphical representation of statistics, * indicates $P < 0.05$, ** $P < 0.01$, and *** $P < 0.001$. Membrane potentials are given without correction for liquid junction potentials. In total, data reported in this paper were

obtained from 136 paired mossy fiber terminal–CA3 pyramidal neuron recordings. 12 of these recordings (Fig. 2g, 10 mM EGTA VC$_{ctrl}$) were already reported in a previous publication[10].

**Reporting summary**. Further information on research design is available in the Nature Research Reporting Summary linked to this article.

## Data availability

A source data file is provided with this paper. Additional binary data files are available from the corresponding authors upon reasonable request.

## Code availability

Analysis routines and code are available from the corresponding authors upon reasonable request.

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

## Acknowledgements

We thank Drs. Carolina Borges-Merjane and Jose Guzman for critically reading the manuscript, and Pablo Castillo for discussions. We are grateful to Alois Schlögl for help with analysis, Florian Marr for excellent technical assistance and cell reconstruction, Christina Altmutter for technical help, Eleftheria Kralli-Beller for manuscript editing, and the Scientific Service Units of IST Austria for support. This project received funding from the European Research Council (ERC) under the European Union's Horizon 2020 research and innovation program (grant agreement No 692692) and the Fond zur Förderung der Wissenschaftlichen Forschung (Z 312-B27, Wittgenstein award), both to P.J.

## Author contributions

D.V. and Y.O. performed the experiments and analyzed the data, P.J. and D.V. conceived the project and wrote the paper. All authors analyzed data and jointly revised the paper.

## Competing interests

The authors declare no competing interests.
