## [Peer Review File · Nature Communications]

Reviewer #1 (Remarks to the Author):

Spatial and temporal accuracy is also required for accurate and unreasonable information transfer between neurons, but a control system capable of controlling the ideal behavior of the entire neuronal network due to excessive signal transfer is needed.

Vandael et al., showed that presynaptic dependent Post-tetanic Potentiation (PTP) in synapses between mossy fiber terminals and postsynaptic CA3 neuron can be regulated by the activity of CA3 neuron. In other words, it was shown that the anti-associative induction, such as a feedback system, can control the expression of the excessive presynaptic activity. Overall, through a very well-designed experimental technique and method, Authors drew accurate and clear results. And this manuscript was reasonably well written including convincing content.

However, there are still a few allegations to fully understand the role of PTP for functional analyses of synaptic plasticity.

Main Issues

1. In a recent study by author (Vandael et al., 2020), it was found that an increase in readily releasable pool (RRP) size is the main cause of PTP. If so, it is reasonable to show whether the occurrence or failure of PTP in various kinds of experiments conducted in this study was due to a change in RRP size or simply as a change in the release probability.
2. When PTP occurred, based on the representative traces shown in this study, the size of the phasic release was increased considerably, but the size of tonic release was unaltered. This result seems to be due to the heterogeneity of RRP. So, do the authors think that PTP is a phenomenon occurring only in a particular pool increase?
3. Is there any relationship between PTP induction and interneuron activity in your preparation ?

Minor Issues

1. In Fig. 1f, are the positions of numbers in y-axis appropriate? The bar graph locates in the negative.
2. One of the traces mentioned in the Fig. 2d legend is said to be "red", but the red trace can not be found in Fig. 2d. Did you refer to the "violet" trace as the "red" trace?
3. In Fig. 3, the trace (a) is indicated that PTP occurred during application of AP5, but can not find it on graph (b and c). If there is trace that matches the graph, it is better to change it.
4. In Fig. 4a-c, the size of EPSC was clearly increased in 4 out of 6 experiments and decreased in the rest of experiments after PTP induction. Are these change in EPSC sizes caused by the changes in RRP?
5. On page 10, line 3, is mGluR2 marked as MGluR2?
6. In Fig. 4h, the shape of PTP is different, compared to the others. The increased EPSCs after PTP induction were maintained for a while. I ask you to add a brief explanation of why this is so.

Reviewer #2 (Remarks to the Author):

In this study Vandael and colleagues examine cooperativity and associativity of hippocampal MF-CA3 short term plasticity (PTP). Importantly, the authors use a technically challenging paired mossy fiber bouton-postsynaptic CA3 pyramidal (MF-CA3) cell experimental setup to unambiguously identify single synaptic inputs which is critical to rigorously probe these features of PTP without the confounds associated with multi-fiber stimulation approaches. As previously hinted at in the

literature MF-CA3 PTP is demonstrated to be input specific and non-cooperative. Most interestingly, the authors reveal a previously undocumented anti-associative property to MF-CA3 PTP whereby coincident spiking activity in the postsynaptic cell during presynaptic tetanization suppresses PTP. Evidence is provided that, mechanistically, associative suppression of PTP proceeds by dendritic R-type (and partially L-type) calcium channel activation with subsequent release of glutamate to activate presynaptic group II mGluRs which are well known to presynaptically depress MF-CA3 synapses. The study is technically excellent with thoughtful and well executed experiments yielding insightful novel findings that are expertly dissected and communicated. Moreover, the newly discovered anti-associative property has significant potential physiological relevance by placing a brake on the MF teacher/detonator synapses that are considered important for memory storage and recall. Overall the study provides important new insight into synaptic plasticity that is sure to be of great interest to the entire neuroscience community. However, the following points should be addressed to improve the manuscript.

Major Points:

- 1) Under basal transmission MF-CA3 synapses are susceptible to depression by group II mGluR activation with exogenously supplied agonist. Do MF-CA3 synapses exhibit depolarization induced suppression of excitation (DSE) due to liberation of glutamate following postsynaptic depolarization to activate R- and L-type VGCCs? While not necessarily physiologically relevant experimental demonstration of MF-CA3 basal transmission DSE would provide strong support for the authors proposed mechanism for suppression of PTP.
- 2) Prior work has provided evidence that glutamate spillover from MF themselves may activate MF group II mGluRs under special circumstances such as lower recording temperatures where glutamate transporter activity is reduced (Scanziani et al., 1997; Min et al., 1998). The current study is performed at room temperature. Does suppression of PTP proceed at physiological temperatures or does the increased transporter activity prevent PTP suppression?
- 3) Mechanistically, presynaptic group II mGluR activation has previously been demonstrated to depress MF bouton calcium transients. Very recent work from the Jonas group elegantly indicates that PTP reflects an increase in the readily releasable pool (RRP) of synaptic vesicles (a "pool engram"). This leads to the question of whether the observed suppression of PTP actually reflects a molecular block of the PTP mechanism (as discussed by the authors, pg 9 2nd pgph) or instead a masking of PTP through a completely alternate molecular target (ie. Inhibition of presynaptic P/Q type VGCCs)?
- 4) In their recently published work, Jonas and colleagues provide unprecedented insight into the kinetics of MF-CA3 PTP and remarkably show that the lifetime of PTP can be extended by absence of presynaptic activity. From a physiological viewpoint it would be important to further probe the parameter space for the time course of the newly discovered anti-associative suppression of PTP. For example, if one refrains from probing PTP by post-tetanus presynaptic stimulation for various intervals to maintain the presynaptic PTP engram pool substrate is there a discrete window during which liberated postsynaptic glutamate can suppress PTP? The answer should depend on whether the retrograde signal does indeed prevent formation of the structural pool engram as suggested or simply masks PTP with temporary reversible presynaptic VGCC inhibition.
- 5) The authors elegantly illustrate that PTP is non-cooperative and synapse specific. However, given the proposed mechanism of a retrograde glutamate signal and the fact that group II mGluRs may be localized distal to presynaptic terminals, it would be of interest to probe whether anti-associative suppression of PTP is synapse specific.

Minor points:

- 1) Many early studies using field potential recordings illustrated robust MF-CA3 PTP. These experiments would be akin to the “free spiking” conditions of the current study which illustrate blunted PTP. Some discussion/comparison to this earlier work would seem appropriate.
- 2) Are the experimental conditions for Fig 2g (0.1 EGTA) VCctrl equivalent to those for the data presented in Fig 1d-f? If so, please elaborate on the dramatically different degrees of PTP. Does it relate to initial release probability differences observed with paired pulse probing stimuli before and after HFS100 (Fig 1 data set) versus train probing stimuli (Fig 2 data set)? It is a little confusing as the 0.1EGTA VCctrl data set is introduced as a new experimental condition on pg 6 1st pgph but appears to be a similar condition to the data provided in Fig1d-f bouton.
- 3) Related to point 2 above, based on the proposed mechanism one would expect a significant difference in PTP magnitude between the following conditions: CCFree vs VCctrl both in 0.1EGTA presented in Fig2g. Do the authors expect that the VC condition is insufficient to fully prevent R-type VGCC activation during HFS100?
- 4) Throughout, “break” should be “brake”.

Reviewer #3 (Remarks to the Author):

In this manuscript, Vandael and colleagues have examined post-tetanic potentiation (PTP) at hippocampal mossy fibre synapses – a form of short-term plasticity that is widely believed to be presynaptically induced and expressed. Using an elegant paired patch-clamp approach to simultaneously stimulate single presynaptic terminals (mossy fibre boutons) and record the corresponding responses from their postsynaptic partners (CA3 pyramidal neurons) in rat hippocampal slices, the authors confirm that PTP induced by high-frequency stimulation (HFS) is synapse-specific and does not require the cooperative activity of multiple inputs for its full expression. The key and novel finding here, however, is that when presynaptic HFS is paired (associated) with postsynaptic activity, PTP is significantly reduced or completely eliminated – a phenomenon the authors refer to as ‘anti-associativity’. After demonstrating that a rise in postsynaptic calcium is required for PTP inhibition to occur, the authors go on to identify R- and L-type voltage-gated channels as the most likely calcium source involved in this anti-associative process. Furthermore, the authors show that PTP can be rescued during associative pairing by either bath application of the selective group II mGluR antagonist LY341495, or by patch-loading an inhibitor of glutamate vesicle refilling into the postsynaptic cell. These findings lead the authors to propose a retrograde signalling mechanism for anti-associative PTP involving the release of glutamate from postsynaptic dendrites and the activation of presynaptic mGluR2/3.

Overall, this is a solid and well-presented study, with appropriate controls, sample sizes, and statistical analyses. The key finding that mossy fibre PTP has anti-associative properties is novel, and reveals an extra layer of complexity to the induction rules governing synaptic plasticity at this synapse. I have no major concerns. However, the authors should address the following minor points related primarily to the clarity of the text and experimental procedures.

Minor points to address:

- It is not clear which of the internal solution recipes was used for the VC experiments presented in Fig. 1. Presumably a K-gluconate based solution containing 10 mM EGTA?

- It appears as though only paired stimuli (at 50 Hz?) were delivered for the recordings in Fig. 1, rather than the 10-stimuli protocol described in the methods and used in the other figures. This different stimulation protocol should be made clear in the methods.

- 'HFS100' should be more clearly defined in the methods and/or its first use in the results section (page 4 paragraph 2). Presumably it is 100 stimuli delivered at 100 Hz?

- In results section 2 (Anti-associativity), it is not clear from the text that presynaptic stimulation was delivered via the MFB tight-seal configuration (rather than field stimulation of the tract). Although this detail is mentioned in the corresponding figure legend, it would be useful to also make it clear at the start of the results text, since both MFB and tract stimulation were described in the preceding results section.

- In the first and third results sections, the authors report the raw pA values of EPSCs before and after PTP induction, whereas in results section 2 the level of PTP is reported as a normalised percentage of baseline. It would be useful to add the percent-of-baseline values for level of PTP for all the experimental groups across all the results sections (in addition to the raw pA values where appropriate).

- Some further details should be given in the methods and/or results for the protocols used when switching to current clamp mode in Figs 2–4. Firstly, was a manual holding current applied to cells in order to maintain a particular resting potential, or was $I=0$ used? Secondly, what are the specifics of the current injection protocol used to deliver HFS100 to the postsynaptic cells (i.e. the magnitude and duration of the currents used to elicit spiking).

- In the methods (page 18), the authors state that “Membrane potentials are given without correction for liquid junction potentials”. However, no membrane potentials are reported in the manuscript. It would be useful to report the resting potential of the postsynaptic CA3 cells recorded in current clamp just prior to the delivery of HFS100 (assuming $I=0$ was used). This could be given in the methods as a simple average value for all recordings (together with any criteria used to exclude cells on the basis of out-of-range resting potentials), or separately for each group of recordings in the figure legends.

- In several instances, the authors have written “to implement a break on mossy fibre detonation” (last sentence of the abstract; page 5 paragraph 2; page 10 paragraph 2). Whereas in one instance they have used the word “brake” in this context (page 6 paragraph 1). I believe that “brake” would be the correct spelling for the intended meaning.

- On page 5 paragraph 3, that authors write “The number of postsynaptic APs observed after presynaptic HFS100 ...”. Should this be “... during presynaptic HFS100 ...”?

- Page 5, paragraph 3 it is written: “... was significantly reduced as compared to control experiments performed in previous work ...”. It should be made clearer that these ‘control experiments’ are (presumably) voltage-clamp recordings.

- Page 7 paragraph 1: “Given that T-type channels suffer from voltage-dependent inactivation ...”. I would suggest changing ‘suffer from’ to ‘exhibit’.

- Page 18 paragraph 1: "12 of these recordings (Fig. 2g, VCnoAPs) were already reported ...". In Fig. 2g, these recordings are labelled as 'VCtrl' rather than 'VCnoAPs'.

- Page 20: "d, Representative traces of control (black) and post HFS100 pairing (red)." 'red' should be 'violet'.

NCOMMS-20-35822, Vandael et al.

We thank the reviewers for their careful reading of our paper, the positive evaluation, and the constructive suggestions.

Point-by-point reply to comments of reviewer #1

Spatial and temporal accuracy is also required for accurate and unreasonable information transfer between neurons, but a control system capable of controlling the ideal behavior of the entire neuronal network due to excessive signal transfer is needed.

Vandael et al., showed that presynaptic dependent Post-tetanic Potentiation (PTP) in synapses between mossy fiber terminals and postsynaptic CA3 neuron can be regulated by the activity of CA3 neuron. In other words, it was shown that the anti-associative induction, such as a feedback system, can control the expression of the excessive presynaptic activity. Overall, through a very well-designed experimental technique and method, Authors drew accurate and clear results. And this manuscript was reasonably well written including convincing content.

We thank the reviewer for several positive statements (“very well-designed”, “accurate and clear results”, “reasonably well written”, “convincing content”).

However, there are still a few allegations to fully understand the role of PTP for functional analyses of synaptic plasticity.

Main Issues

1. In a recent study by author (Vandael et al., 2020), it was found that an increase in readily releasable pool (RRP) size is the main cause of PTP. If so, it is reasonable to show whether the occurrence or failure of PTP in various kinds of experiments conducted in this study was due to a change in RRP size or simply as a change in the release probability.

We thank the reviewer for this suggestion. We have generated cumulative EPSC plots for the two conditions shown in Fig. 2a–c and Fig. 2d–f, and included the results as a Supplementary Figure (Supplementary Fig. 1). The results suggest that both PTP and its modulation converge on changes in RRP size.

2. When PTP occurred, based on the representative traces shown in this study, the size of the phasic release was increased considerably, but the size of tonic release was unaltered. This result seems to be due to the heterogeneity of RRP. So, do the authors think that PTP is a phenomenon occurring only in a particular pool increase?

We observed little change in the frequency of spontaneous (“tonic”) EPSCs, as expected by the reviewer (Fig. 1 for reviewers). We now mention this on p. 6, top of the revised manuscript. However, we need to consider that a single CA3 pyramidal neuron receives input from ~50 mossy fiber terminals, while only a single input is stimulated. Thus, any increase in spontaneous EPSC frequency will be difficult to detect.

3. Is there any relationship between PTP induction and interneuron activity in your preparation ?

This is a very good question. We have been avoiding gabazine or other GABAA receptor blockers, to preserve the natural conditions as much as possible. In Vyleta et al., 2016, we tested the effects of gabazine, and found no detectable differences between responses in the absence and the presence of gabazine. We are planning to expand this analysis in a future study,

Minor Issues

1. In Fig. 1f, are the positions of numbers in y-axis appropriate? The bar graph locates in the negative.

We have amended the figure as suggested.

2. One of the traces mentioned in the Fig. 2d legend is said to be “red”, but the red trace can not be found in Fig, 2d. Did you refer to the “violet” trace as the “red” trace?

We apologize for the mistake. We have corrected the legend as suggested.

3. In Fig. 3, the trace (a) is indicated that PTP occurred during application of AP5, but can not find it on graph (b and c). If there is trace that matches the graph, it is better to change it.

We have replaced the trace by a more representative one, as suggested.

4. In Fig. 4a-c, the size of EPSC was clearly increased in 4 out of 6 experiments and decreased in the rest of experiments after PTP induction. Are these change in EPSC sizes caused by the changes in RRP?

We have analyzed pool parameters for individual experiments, as requested. As expected by the reviewer, we have seen, a clear increase in the RRP in 3 out of 6 pairs (Fig. 2 for reviewers). However, we want to point out that the effect of HFS_{pre/post} did not reach statistical significance, neither for EPSC₁ amplitude (Fig. 4c), nor for RRP size (Fig. 2 for reviewers).

5. On page 10, line 3, is mGluR2 marked as MGluR2?

MGluR2 was capitalized, because it was at the beginning of the sentence. However, we agree this might be misleading, and changed the label as requested by the reviewer.

6. In Fig. 4h, the shape of PTP is different, compared to the others. The increased EPSCs after PTP induction were maintained for a while. I ask you to add a brief explanation of why this is so.

We thank the reviewer for sharply detecting this difference. We have tested the differences for statistical significance by two-way ANOVA and found no statistically significant interaction ($P = 0.91$). We have added a sentence to the figure legend, which hopefully provides some clarification (p. 25, top). It is possible that some activity-

induced suppression of PTP is remaining in the presence of concanamycin, but this residual suppression is more short-lasting.

Point-by-point reply to comments of reviewer #2

In this study Vandael and colleagues examine cooperativity and associativity of hippocampal MF-CA3 short term plasticity (PTP). Importantly, the authors use a technically challenging paired mossy fiber bouton-postsynaptic CA3 pyramidal (MF-CA3) cell experimental setup to unambiguously identify single synaptic inputs which is critical to rigorously probe these features of PTP without the confounds associated with multi-fiber stimulation approaches. As previously hinted at in the literature MF-CA3 PTP is demonstrated to be input specific and non-cooperative. Most interestingly, the authors reveal a previously undocumented anti-associative property to MF-CA3 PTP whereby coincident spiking activity in the postsynaptic cell during presynaptic tetanization suppresses PTP. Evidence is provided that, mechanistically, associative suppression of PTP proceeds by dendritic R-type (and partially L-type) calcium channel activation with subsequent release of glutamate to activate presynaptic group II mGluRs which are well known to presynaptically depress MF-CA3 synapses. The study is technically excellent with thoughtful and well executed experiments yielding insightful novel findings that are expertly dissected and communicated. Moreover, the newly discovered anti-associative property has significant potential physiological relevance by placing a brake on the MF teacher/detonator synapses that are considered important for memory storage and recall. Overall the study provides important new insight into synaptic plasticity that is sure to be of great interest to the entire neuroscience community. However, the following points should be addressed to improve the manuscript.

We thank the reviewer for his / her positive statements (“technically excellent”, “thoughtful and well executed experiments”, “insightful novel findings”, “expertly dissected”, “important new insight”, “great interest to the entire neuroscience community”).

Major Points:

1) Under basal transmission MF-CA3 synapses are susceptible to depression by group II mGluR activation with exogenously supplied agonist. Do MF-CA3 synapses exhibit depolarization induced suppression of excitation (DSE) due to liberation of glutamate following postsynaptic depolarization to activate R- and L-type VGCCs? While not necessarily physiologically relevant experimental demonstration of MF-CA3 basal transmission DSE would provide strong support for the authors proposed mechanism for suppression of PTP.

This is an excellent suggestion. To test for DSE, we have applied trains of eight 100-ms voltage-clamp stimuli to 0 mV (900-ms interval) in the postsynaptic CA3 cell and monitored evoked EPSCs before, during, and after stimulation (Fig. 3 for reviewers). Indeed, we found that the EPSC amplitude was reduced after postsynaptic stimulation. Furthermore, LY341495 abolished this reduction, consistent with our hypothesis that retrograde glutamate signaling may be involved. However, a major complication is that the kinetics of depression does not fit to that typically reported for DSI or DSE, but seemed to be more long-lasting (Fig. 3a–c for reviewers). Thus, we think the phenomenon may be distinct from DSE. Given these complications, we strongly prefer not to include these results in the current manuscript.

2) Prior work has provided evidence that glutamate spillover from MF themselves may activate MF group II mGluRs under special circumstances such as lower recording temperatures where glutamate transporter activity is reduced (Scanziani et al., 1997; Min et al., 1998). The current study is performed at room temperature. Does

suppression of PTP proceed at physiological temperatures or does the increased transporter activity prevent PTP suppression?

We thank the reviewer for this comment. We have performed additional experiments at near-physiological temperature. The results indicate that the basic phenomenon is preserved, although the magnitude of the effect may be reduced. We have added a Supplementary Figure to the manuscript to document this important point (Supplementary Fig. 2). It is possible that at room temperature the AP that triggers dendritic release is longer, or the uptake of dendritically released glutamate is prolonged (as suggested by the reviewer). Both factors are expected to enhance the retrograde signaling mechanism.

3) Mechanistically, presynaptic group II mGluR activation has previously been demonstrated to depress MF bouton calcium transients. Very recent work from the Jonas group elegantly indicates that PTP reflects an increase in the readily releasable pool (RRP) of synaptic vesicles (a “pool engram”). This leads to the question of whether the observed suppression of PTP actually reflects a molecular block of the PTP mechanism (as discussed by the authors, pg 9 2nd pgph) or instead a masking of PTP through a completely alternate molecular target (ie. Inhibition of presynaptic P/Q type VGCCs)?

We thank the reviewer for this comment. We have included additional analysis to distinguish between changes in RRP size and release probability, confirming that the transsynaptic modulation of PTP is largely mediated via changes in RRP size, as is PTP itself (Supplementary Fig. 3). The results corroborate the conclusion that both PTP and its modulation are mainly mediated by changes in pool size. However, we cannot exclude that changes in release probability also contribute to the suppression. We have added a caveat sentence to cover this point (p. 11, top of the revised manuscript).

4) In their recently published work, Jonas and colleagues provide unprecedented insight into the kinetics of MF-CA3 PTP and remarkably show that the lifetime of PTP can be extended by absence of presynaptic activity. From a physiological viewpoint it would be important to further probe the parameter space for the time course of the newly discovered anti-associative suppression of PTP. For example, if one refrains from probing PTP by post-tetanus presynaptic stimulation for various intervals to maintain the presynaptic PTP engram pool substrate is there a discrete window during which liberated postsynaptic glutamate can suppress PTP? The answer should depend on whether the retrograde signal does indeed prevent formation of the structural pool engram as suggested or simply masks PTP with temporary reversible presynaptic VGCC inhibition.

We thank the reviewer for this excellent comment. We have added additional analysis to distinguish between changes in RRP size and release probability, confirming that the transsynaptic modulation of PTP is largely mediated via changes in RRP size, as is PTP itself (Supplementary Fig. 1). We agree with the reviewer that it would be nice to systematically vary the relative timing of pre- and postsynaptic stimulation. However, we think that this would require huge effort that may well form the basis of a separate paper. As the reviewer knows, paired mossy fiber terminal CA3 pyramidal neuron recordings are technically extremely challenging, and the number of experiments in the present paper is already at 133.

5) The authors elegantly illustrate that PTP is non-cooperative and synapse specific. However, given the proposed mechanism of a retrograde glutamate signal and the fact that groupII mGluRs may be localized distal to presynaptic terminals, it would be of interest to probe whether anti-associative suppression of PTP is synapse specific.

We thank the reviewer for this comment. The ideal way to tackle this problem would be by controlled paired recordings of multiple terminals converging on the same postsynaptic CA3 pyramidal neuron. We are planning to further investigate this in a future manuscript. We have added a sentence to mention this in the revised paper (p. 10, top of the revised manuscript).

Minor points:

1) Many early studies using field potential recordings illustrated robust MF-CA3 PTP. These experiments would be akin to the “free spiking” conditions of the current study which illustrate blunted PTP. Some discussion/comparison to this earlier work would seem appropriate.

This is an excellent point. One possibility is that the endogenous Ca^{2+} buffering of CA3 pyramidal neurons is higher than that corresponding to 0.1 mM EGTA. We have introduced a paragraph to discuss this issue, also referring to several published papers (Salin et al., 1996; Huang and Kandel, 1996; p. 12, top of the revised manuscript).

2) Are the experimental conditions for Fig 2g (0.1 EGTA) VCctrl equivalent to those for the data presented in Fig 1d-f? If so, please elaborate on the dramatically different degrees of PTP. Does it relate to initial release probability differences observed with paired pulse probing stimuli before and after HFS100 (Fig 1 data set) versus train probing stimuli (Fig 2 data set)? It is a little confusing as the 0.1EGTA VCctrl data set is introduced as a new experimental condition on pg 6 1st pgph but appears to be a similar condition to the data provided in Fig1d-f bouton.

We thank the reviewer for pointing this out. As the experiments in Fig. 1 were performed with 10 mM EGTA in the postsynaptic pipette, there is actually no discrepancy. We have added this important information in the legend of Fig. 1, as well as in the main text (p. 4, top of the revised manuscript).

3) Related to point 2 above, based on the proposed mechanism one would expect a significant difference in PTP magnitude between the following conditions: CCfree vs VCctrl both in 0.1EGTA presented in Fig2g. Do the authors expect that the VC condition is insufficient to fully prevent R-type VGCC activation during HFS100?

This is exactly our interpretation. We have added a sentence to the manuscript to better explain this (p. 10, bottom of the revised paper).

4) Throughout, “break” should be “brake”.

The reviewer is absolutely right. We have corrected this error throughout.

Point-by-point reply to comments of reviewer #3

In this manuscript, Vandael and colleagues have examined post-tetanic potentiation (PTP) at hippocampal mossy fibre synapses – a form of short-term plasticity that is widely believed to be presynaptically induced and expressed. Using an elegant paired patch-clamp approach to simultaneously stimulate single presynaptic terminals (mossy fibre boutons) and record the corresponding responses from their postsynaptic partners (CA3 pyramidal neurons) in rat hippocampal slices, the authors confirm that PTP induced by high-frequency stimulation (HFS) is synapse-specific and does not require the cooperative activity of multiple inputs for its full expression. The key and novel finding here, however, is that when presynaptic HFS is paired (associated) with postsynaptic activity, PTP is significantly reduced or completely eliminated – a phenomenon the authors refer to as ‘anti-associativity’. After demonstrating that a rise in postsynaptic calcium is required for PTP inhibition to occur, the authors go on to identify R- and L-type voltage-gated channels as the most likely calcium source involved in this anti-associative process. Furthermore, the authors show that PTP can be rescued during associative pairing by either bath application of the selective group II mGluR antagonist LY341495, or by patch-loading an inhibitor of glutamate vesicle refilling into the postsynaptic cell. These findings lead the authors to propose a retrograde signalling mechanism for anti-associative PTP involving the release of glutamate from postsynaptic dendrites and the activation of presynaptic mGluR2/3.

Overall, this is a solid and well-presented study, with appropriate controls, sample sizes, and statistical analyses. The key finding that mossy fibre PTP has anti-associative properties is novel, and reveals an extra layer of complexity to the induction rules governing synaptic plasticity at this synapse. I have no major concerns. However, the authors should address the following minor points related primarily to the clarity of the text and experimental procedures.

We thank the reviewer for his / her positive statements (“solid”, “well-presented”, “novel”, “no major concerns”).

Minor points to address:

- It is not clear which of the internal solution recipes was used for the VC experiments presented in Fig. 1. Presumably a K-gluconate based solution containing 10 mM EGTA?

We have clarified this in the revised paper (p. 4, top of the revised manuscript).

- It appears as though only paired stimuli (at 50 Hz?) were delivered for the recordings in Fig. 1, rather than the 10-stimuli protocol described in the methods and used in the other figures. This different stimulation protocol should be made clear in the methods.

We have added the relevant information in the paper (p. 18, bottom of the revised manuscript).

- ‘HFS100’ should be more clearly defined in the methods and/or its first use in the results section (page 4 paragraph 2). Presumably it is 100 stimuli delivered at 100 Hz?

We have better defined the paradigm, as requested by the reviewer (p. 4, top of the revised manuscript).

- In results section 2 (Anti-associativity), it is not clear from the text that presynaptic stimulation was delivered via the MFB tight-seal configuration (rather than field stimulation of the tract). Although this detail is mentioned

in the corresponding figure legend, it would be useful to also make it clear at the start of the results text, since both MFB and tract stimulation were described in the preceding results section.

We now mention the use of tight-seal cell-attached stimulation, as requested by the reviewer (p. 5, top of the revised manuscript).

- In the first and third results sections, the authors report the raw pA values of EPSCs before and after PTP induction, whereas in results section 2 the level of PTP is reported as a normalised percentage of baseline. It would be useful to add the percent-of-baseline values for level of PTP for all the experimental groups across all the results sections (in addition to the raw pA values where appropriate).

We have added the PTP percentage values throughout, as suggested by the reviewer (p. 4 to 8 of the revised manuscript). As we continue to think that the non-normalized amplitude values are also informative, we in several cases now report both numbers in parallel. Moreover, we have included a summary bar graph of PTP percentage values, including systematic multiple comparison analysis of all pharmacological manipulations (Supplementary Fig. 4).

- Some further details should be given in the methods and/or results for the protocols used when switching to current clamp mode in Figs 2–4. Firstly, was a manual holding current applied to cells in order to maintain a particular resting potential, or was I=0 used? Secondly, what are the specifics of the current injection protocol used to deliver HFS100 to the postsynaptic cells (i.e. the magnitude and duration of the currents used to elicit spiking)

We have added information about the holding current and the specific HFS protocol (pulse amplitude and duration) to the methods section, as requested by the reviewer (p. 19, center of the revised manuscript).

- In the methods (page 18), the authors state that “Membrane potentials are given without correction for liquid junction potentials”. However, no membrane potentials are reported in the manuscript. It would be useful to report the resting potential of the postsynaptic CA3 cells recorded in current clamp just prior to the delivery of HFS100 (assuming I=0 was used). This could be given in the methods as a simple average value for all recordings (together with any criteria used to exclude cells on the basis of out-of-range resting potentials), or separately for each group of recordings in the figure legends.

We have added information about the resting membrane potential in the methods section, as requested by the reviewer (p. 19, center of the revised manuscript).

- In several instances, the authors have written “to implement a break on mossy fibre detonation” (last sentence of the abstract; page 5 paragraph 2; page 10 paragraph 2). Whereas in one instance they have used the word “brake” in this context (page 6 paragraph 1). I believe that “brake” would be the correct spelling for the intended meaning.

We apologize for this mistake. We have corrected the spelling as requested.

- On page 5 paragraph 3, that authors write “The number of postsynaptic APs observed after presynaptic HFS100 ...”. Should this be “... during presynaptic HFS100 ...”?

We have corrected this inaccuracy as requested (p. 6, top of the revised manuscript).

- Page 5, paragraph 3 it is written: "... was significantly reduced as compared to control experiments performed in previous work ...". It should be made clearer that these 'control experiments' are (presumably) voltage-clamp recordings.

We have added this important information as requested (p. 6, center of the revised manuscript).

- Page 7 paragraph 1: "Given that T-type channels suffer from voltage-dependent inactivation ...". I would suggest changing 'suffer from' to 'exhibit'.

We admit that "suffer" sounds subjective, and made the necessary amendment (p. 8, top of the revised manuscript).

- Page 18 paragraph 1: "12 of these recordings (Fig. 2g, VCnoAPs) were already reported ...". In Fig. 2g, these recordings are labelled as 'VCCtrl' rather than 'VCnoAPs'.

We have changed the text as suggested.

- Page 20: "d, Representative traces of control (black) and post HFS100 pairing (red)". 'red' should be 'violet'.

We have corrected the label description as requested.

We hope that, after these revisions, which include new experiments, additional analysis, and requested modification of the text, the manuscript will be suitable for publication in *Nature Communications*.

Once again, we thank all reviewers for helping us to further improve our paper.

Figure 1 for reviewers | Spontaneous EPSCs before and after HFS in 0.1 mM and 10 mM EGTA.

a, b, Spontaneous EPSCs before and after HFS₁₀₀ in 0.1 mM EGTA (a) and 10 mM EGTA (b).

c, d, Summary bar graph of spontaneous EPSC frequency (c) and peak amplitude (d).

EPSCs were detected using a template-fit algorithm for an 8-s time interval before and a similar time interval after HFS₁₀₀. Note that EPSC frequency was not significantly different before and after HFS₁₀₀, in both 0.1 mM and 10 mM EGTA.

Figure 2 for reviewers | Pool analysis of individual experiments in AM251, as in Fig. 4a–c.

Cumulative EPSC peak amplitude was plotted against stimulus number, and data were analyzed by linear regression of the last 4 data points (Vandael et al., 2020). The size of the RRP was determined as the intersection of the regression line with the

ordinate, P_r was measured as the ratio of the first EPSC amplitude over RRP size, and refilling rate was obtained from the slope of the regression line.

Figure 3 for reviewers | Depolarization-induced suppression of excitation (DSE) induced by postsynaptic stimulation.

a-c, Plot of EPSC₁, EPSC₂, and EPSC₃ against experimental time. Vertical dashed line indicates the time point of the stimulation protocol (eight 100-ms voltage clamp pulses to 0 mV, 900-ms interpulse interval). Horizontal dashed line indicates baseline.

d, Plot of EPSC₁ against experimental time in the presence of the mGluR2/3 antagonist LY341495. Vertical black dashed line indicates the time point of the DSE protocol. Vertical gray dashed line shows the time point of a subsequent HFS₁₀₀ protocol.

e-g, Summary bar graphs of DSE (e), DSE in the presence of LY341495 (f), and HFS₁₀₀-induced PTP (g).

For the quantification of DSE, we compared the average value of 100 s before depolarization with that of 180 s after depolarization. For the quantification of PTP, we compared the average value of 100 s pre HFS₁₀₀ to that 20 s post HFS₁₀₀.

Reviewer #1 (Remarks to the Author):

The authors prepared adequately satisfactory answers to all questions.

Reviewer #2 (Remarks to the Author):

I commend the authors on an excellent manuscript. Their revisions and responses fully address the points raised in the original review and have strengthened the manuscript. I have no further concerns to be addressed.

Ken Pelkey